# A Meta-Analysis of Functional Communication Training for Young Children with ASD and Challenging Behavior in Natural Settings

**DOI:** 10.3390/bs15121688

**Published:** 2025-12-05

**Authors:** Kwang-Sun Cho Blair, Eun-Young Park, Madeline R. Risse

**Affiliations:** 1Applied Behavior Analysis Program, Department of Child and Family Studies, University of South Florida, Tampa, FL 33612, USA; kwangsun@usf.edu (K.-S.C.B.); mrisse@usf.edu (M.R.R.); 2Department of Secondary Special Education, Jeonju University, Jeonju 55069, Republic of Korea

**Keywords:** functional communication training, challenging behavior, young children, autism spectrum disorder, meta-analysis

## Abstract

This meta-analysis synthesized 34 published single-case design studies on functional communication training (FCT) for young children with autism spectrum disorder (ASD). A systematic search of electronic databases and reference lists identified studies published between 1996 and 2021 involving 79 children with ASD aged 2 to 8. Quality evaluation using What Works Clearinghouse standards revealed that only 29.4% of studies met standards with or without reservations, primarily due to insufficient data points per phase. Most studies were conducted in home or school settings with therapists/researchers as primary implementers, followed by parents or caregivers. Low reporting rates were found for preference assessment, treatment fidelity, social validity, and maintenance and generalization effects. Overall, FCT demonstrated large effects for reducing challenging behavior (Tau-BC = 0.97) and moderate-to-large effects for increasing replacement behavior (Tau-BC = 0.78). Moderator analyses revealed significantly larger effect sizes in school versus home settings (*p* < 0.05). These findings further support FCT as an evidence-based practice for young children with ASD, although methodological rigor must be improved. Future research should systematically evaluate maintenance and generalization effects, develop effective parent training and support strategies, and report intervention dosage parameters to strengthen the evidence base and guide clinical implementation.

## 1. Introduction

Autism spectrum disorder (ASD) is a complex neurodevelopmental condition characterized by ongoing challenges in social communication and interaction, along with restricted and repetitive behavioral patterns, interests, or activities ([65]). A defining characteristic of ASD is the wide variation in how symptoms present. Individuals with ASD may demonstrate different levels of intellectual ability, sensory processing, and adaptive skills, leading to various behavioral manifestations that make designing and implementing interventions challenging ([62]; [107]). Among this diversity, challenging or maladaptive behaviors pose especially significant concerns for families and practitioners, as they impact individuals’ everyday functioning, learning experiences, and overall well-being ([64]; [29]). These behaviors include, but are not limited to, aggression, self-injurious behavior, tantrums, repetitive movements, elopement, sleep and eating difficulties, and sensory sensitivities, among others. The literature indicates that many children with ASD exhibit high levels of tantrums, aggression, and self-injury ([74]). For example, in a North American sample (*N* = 1380), [58] ([58]) found that approximately 70% of children with ASD between the ages of 4 and 14 years exhibited severe tantrums during their development, with 49% engaging in aggression toward their parents and 49% in aggression toward non-caregivers. In a population-based surveillance study with a large sample of children with ASD in the United States, [96] ([96]) found that the prevalence of self-injurious behavior in ASD averaged 27.7% over three surveillance years.

Given the central diagnostic features of ASD, specifically, persistent deficits in social communication, understanding the relationship between communication skill deficits and challenging behavior has received substantial research attention ([28]). While this relationship is not unique to ASD, as typically developing children also show associations between communication difficulties and problem behaviors ([25]), the question remains particularly relevant to individuals with ASD. Some evidence suggests that children with more developed communication skills tend to exhibit less challenging behavior ([7]; [108]). However, these relationships often lack robustness when additional variables are controlled ([58]), indicating that challenging behavior is likely due to multiple interacting factors rather than communication deficits alone. Nevertheless, many studies have demonstrated that teaching new alternative communicative behaviors through functional communication training (FCT) intervention leads to reductions in challenging behavior ([27]; [75]), suggesting that such behaviors may serve communicative functions ([28]).

The FCT intervention represents one of the most widely validated approaches for addressing challenging behavior that serves communicative functions in individuals with ASD. The success of FCT depends on two critical components: (a) accurately identifying the antecedents and maintaining consequences of challenging behavior through a functional behavior assessment process and (b) teaching an alternative FCR that serves the same function while being more socially acceptable and efficient ([28]). In addition to a large number of single-case design (SCD) studies, the robust evidence base for FCT is supported by several narrative, systematic, and meta-analytic reviews examining the FCT intervention outcomes for individuals with disabilities (e.g., [1]; [17]; [28]; [33]; [41]; [52]; [66]; [68]; [77]; [79]; [111]). These review studies indicate that FCT effectiveness may not solely be determined by the procedural components (i.e., identifying the behavioral functions and teaching alterative FCRs); individual characteristics (e.g., age, communication mode, and disability type), implementation variables (e.g., treatment fidelity and setting characteristics), and intervention components (e.g., FCT alone versus FCT combined with additional strategies) may serve as important moderators of treatment outcomes ([41]; [52]; [17]).

Existing meta-analytic reviews report moderate-to-large effects of FCT on reducing challenging behavior and increasing functional communication response (FCR), particularly for behaviors maintained by social reinforcement, such as access to tangibles or attention and escape from demands ([41]), and when the mode of communication is vocal ([52]). The reviews indicate that FCT tend to produce better outcomes for preschool-aged children with disabilities ([17]) and individuals with ASD than for those with intellectual disabilities ([52]). However, these findings require careful interpretation, as most studies included in meta-analyses did not report intellectual assessments ([52] ([52]), limiting our understanding of whether FCT effectiveness differs between individuals with ASD alone versus those with co-occurring intellectual disabilities (IDs). Given that approximately 30% of individuals with ASD present with comorbid IDs ([102]), and that cognitive abilities may influence communicative skill acquisition, the failure to differentiate these populations limits our understanding of how individual characteristics moderate FCT outcomes.

The role of natural implementers, such as parents, teachers, and paraprofessionals, is another critical factor in intervention success. Evidence indicates that FCT can be effectively implemented by caregivers with sufficient training and support, maintaining treatment gains over time ([77]; [41]). However, researchers remain the most frequent implementers in published studies (approximately 73%), whereas parents or caregivers served as implementers in only about 10% of the studies examined ([41]). Moreover, treatment fidelity data are often underreported for parent- or caregiver-implemented FCT ([41]). Similarly, in reviewing 12 studies that involved school personnel implementing FCT with children with disabilities, [1] ([1]) found that only 50% of studies included a treatment fidelity measure, highlighting the need for more systematic examination of implementation quality and generalization in natural contexts.

To assess the quality of the studies included in their analyses, researchers have used multiple evaluative frameworks. The What Works Clearinghouse (WWC) Design Standards ([112]) focus on evaluating the internal validity of a study, whereas the quality indicators suggested by [54] ([54]) and [86] ([86]) focus on assessing methodological rigor across multiple dimensions, such as the quality of participant and setting descriptions, dependent and independent variables, baseline, internal and external validity, and social validity. [86] ([86]) further expand these indicators to include inter-observer agreement, Kappa statistics, fidelity, use of blind raters, and assessment of generalization and maintenance effects.

The meta-analytic reviews of FCT studies discussed above provided information on the overall characteristics of studies on FCT, the effects of FCT on challenging behavior and FCR, and potential moderators of these effects, such as age, diagnosis, setting, and interventionist. Although the existing meta-analyses provide valuable information on the current status of FCT literature on children with disabilities, the field would benefit from further identification of potential moderators to understand how those variables may influence outcomes of FCT for young children, particularly children with ASD, and to inform practice and future research. Prior literature suggests that treatment fidelity ([71]), social validity ([10]), and implementer training ([87]) are associated with better outcomes of behavioral interventions for children with ASD. However, these variables have not been examined as potential moderators in the FCT literature.

Therefore, the current study aimed to extend the meta-analytic literature on FCT by focusing on SCD studies involving young children with ASD. Specifically, the study sought to (a) identify varying characteristics of the FCT studies conducted exclusively with young children with ASD, (b) evaluate the methodological quality of the studies, (c) determine the magnitude of FCT effects, and (d) identify potential moderators that may influence outcomes of FCT for young children with ASD, including FCT components, outcome types, implementers, and fidelity and social validity assessments.

## 2. Method

### 2.1. Literature Search and Eligibility Criteria

This literature review searched Academic Search Premier, PsycINFO, Web of Science, and Education Source using combinations of the following terms in full text: “functional communication training”, “behavior”, and “autism” or ‘PDD-NOS’. To ensure the quality and completeness of the electronic search, the research questions were structured according to the PICO (population, intervention, comparison, and outcome) framework ([89]). In this review, the population (P) was children ASD, intervention (I) was FCT, and the outcome (O) was reducing problem behaviors. Because the included studies primarily used SCDs, the comparison component (C) was conceptualized in terms of study design rather than traditional control groups. Based on these PICO components, search terms were developed using both controlled vocabulary (e.g., autism spectrum disorder) and free-text keywords (e.g., “autism”, “PDD-NOS.”, “functional communication training”, and “FCT”, “challenging behavior”, and “problem behavior”). The Boolean operators were used to combine search terms appropriately (e.g., “autism” OR “PDD-NOS”) and to connect the main concepts (“autism” AND “functional communication training” AND “behavior”).

Following general recommendations for systematic reviews ([98]), we searched four major databases covering the fields of psychology, education, and behavioral science. Databases were selected based on their relevance to special education and applied behavior analysis research, and the inclusion of peer-reviewed journals indexed in these databases ensured the quality of the retrieved studies. The database search was conducted through 1 March 2022, with no restrictions on publication date. The search process concluded when literature saturation was confirmed, indicated by the absence of newly identified relevant studies in subsequent searches.

The initial database search resulted in 1645 articles (456 from Academic Search Premier, 275 from PsychINFO, 765 from Web of Science, and 149 from Education Source). An additional 130 articles were identified thorough ancestral searches of key review papers ([28]; [41]; [52]; [68]; [79]). After removing 300 duplicate records, 1475 articles remained for screening. Studies were included if they met the following eligibility criteria: (a) involved children with ASD or Pervasive Developmental Disorder-Not Otherwise Specified (PDD-NOS); (b) included only children aged 1–8 years; (c) implemented an FCT intervention; (d) targeted the reduction of challenging behavior; (e) employed an SCD capable of demonstrating a functional relation between the independent and the dependent variables (e.g., alternating treatments, multiple baseline, and reversal designs); (f) provided graphical data suitable for effect size calculation; (g) conducted in natural settings (i.e., home, school, or community settings); (h) were published in peer-reviewed journals prior to March 2022; and (i) were written in English.

### 2.2. Article Selection Procedures

The screening of the titles and abstracts resulted in the exclusion of 1321 studies that included unrelated terms, such as X-linked gene, synthesis, systematic review, validating, and correlation, or that focused on caregiver or therapist behavior, rather than child behavior. To ensure the reliability of the inclusion and exclusion criteria, we (second and third authors) independently conducted the literature selection process. Any disagreements between reviewers were resolved through consensus. For transparency and reproducibility, Excel was used to organize the literature systematically, remove duplicates, and document each stage of the review. Data was stored in cloud-based files to maintain traceability throughout the review process.

After screening, 154 articles remained to undergo full-text review. The full-text reviews excluded 123 articles due to the articles (a) involving children with disabilities other than ASD (e.g., intellectual disabilities and Down syndrome), (b) including participants over 8 years old, (c) using behavioral procedures other than FCT to improve behavior or social communication skills, (d) not focusing on challenging behavior outcomes, (e) not employing an appropriate SCD, (f) lacking graphical data to calculate effect sizes, or (g) including clinical setting. We independently reviewed the excluded studies during the title, abstract, and full-text screening phases to confirm that they did not meet the inclusion criteria. Any discrepancies were discussed until full consensus was reached. This left 31 studies for in-depth review (see Table A1 in Appendix A for a representative sample of 14 excluded studies). Additional ancestral searches were conducted by screening the reference lists of the 31 included studies. This yielded an additional three articles that met the inclusion criteria, bringing the total to 34 articles for the meta-analysis (see Figure 1 for the PRISMA diagram). The included studies spanned from 1999 to 2021, with relatively consistent publication rates across this period (approximately 1–3 studies per year).

When finalizing the 34 studies for meta-analysis, we independently reviewed the full text of each article and discussed whether the articles met the inclusion criteria, achieving 100% agreement. For coding, an independent coder, a graduate student in applied behavior analysis, trained in the procedures, independently coded a random sample of 30% of the articles (*n* = 10). We calculated the inter-coder agreements between the second or third author and the independent coder for each coding variable by dividing the number of agreements by the number of possible agreements, expressed as a percentage. The initial agreement ranged from 90 to 100% across variables, and all discrepancies were resolved through discussion until 100% consensus was achieved.

### 2.3. Extraction of Graphical Data

We extracted data from line graphs using the Digtizelt version 2.2 digitizer software ([6]), which digitizes graphical data and exports it to Excel spreadsheets for further analysis. The software is considered reliable and valid for digitizing graphical data from SCD studies ([85]). Data extraction procedures varied by experimental design. The procedures for data extraction depended on the type of experimental design. In multiple baseline designs, data from both the baseline and treatment phases were retrieved. For reversal designs, values were extracted from the baseline and treatment phase of all reversals, not just the initial phases.

### 2.4. Variable Coding

#### 2.4.1. Study Characteristics

Table 1 presents the specific coding variables within each category of study characteristics. To examine the child participant characteristics, we coded (a) the number of child participants, (b) age (in months), (c) diagnosis, and (d) target communication mode. For diagnosis, in addition to ASD, other comorbid conditions were coded when this information was available. The communication mode was coded as vocal, picture, sign, or voice output communication aid (VOCA). To examine other study characteristics, specifically those related to the FTC features, we coded (a) target challenging behavior, (b) function of challenging behavior, (c) setting, (d) preference assessment, (e) FCT components, (f) implementer, and (g) implementer training (length of initial training and frequency of coaching or feedback).

For target challenging behavior, we coded (a) aggression (e.g., hitting, kicking, scratching, and pinching), (b) disruption (e.g., throwing objects, climbing on furniture, crawling under tables, and making noises), (c) property destruction, (d) tantrums, (e) self-injury, (f) stereotypy and elopement, (g) crying, (h) screaming, and (i) chin grinding. The function of challenging behavior was coded as social (attention, tangible, or escape) or automatic (non-social). The setting was coded as home, school, or community. Preference assessment was coded by method (indirect, trial-based, or free-operant). The FCT component was coded as either FCT alone or FCT plus other interventions, such as noncontingent reinforcement (NCR), extinction, and delay-to-reinforcement (schedule thinning).

Implementers were coded as parents (mother)/caregiver, teacher, therapist, para-educator, consultant, or researcher. For [70] ([70]), the mother was coded as the implementer based on conducting FCT at home, although generalization data were collected in the school setting. In [32] ([32]), the implementers were mother and researcher, with one child trained by the mother and one child trained by the researcher. Implementer training was coded for presence (yes/no). Training length was coded as <1 hr or ≥1 hr. Frequency of coaching or feedback was coded as daily or weekly. One article ([91]) was coded as ‘two sessions’ because only two coaching or feedback sessions were provided during intervention, rather than daily or weekly. Dependent variables were categorized as challenging behavior or replacement behavior (e.g., compliance, task completion, engagement, and FCR).

#### 2.4.2. Methodological Quality Coding

We evaluated the methodological quality of the studies using the WWC SCD Standards ([112]), which consists of five criteria: (a) availability of graphical data, (b) systematic manipulation of the independent variable, (c) inter-observer agreement (IOA), (d) residual treatment effects, and (e) evidence of effect over time and sufficient data points per phase. Specifically, we applied the WWC SCD Review Protocol Checklist, which includes items such as ‘Is the data presented in a way that allows for visual analysis?’ and ‘Was the independent variable systematically manipulated?’ Each criterion was assessed using a yes/no format. All 34 studies met the first and second criteria.

However, 13 studies failed to meet the third criterion (IOA), and thus were not further evaluated on the remaining criteria. The remaining 21 studies were evaluated on all subsequent criteria to determine whether they meet WWC SCD Standards with or without reservation, or do not meet WWC SCD Standards. We also examined four quality indicators proposed by [54] ([54]) and [86] ([86]): (a) treatment fidelity, (b) social validity, (c) maintenance, and (d) generalization. We coded each indicator dichotomously (yes/no).

### 2.5. Effect Size Calculations

The effect size estimation involved calculating individual Tau-BC indices followed by effect size aggregation. We aggregated the individual Tau-BC effect sizes using a three-level multilevel meta-analysis in SAS Proc MIXED (SAS version 9.4). This model accounted for the hierarchical structure of SCD data, where multiple effect sizes (cases) were nested within studies. The model included a repeated statement grouped by case within study to address potential heteroscedasticity across cases. We used the Kenward–Roger degrees of freedom method to improve small-sample inference accuracy, as recommended for multilevel modeling of SCD studies ([35]).

### 2.6. Moderator Analysis

The potential moderators examined in this study included (a) FCT components (FCT-only vs. FCT plus additional component), (b) outcome (challenging behavior vs. replacement behavior), (c) setting (home, school, or community), (d) implementer (parent, therapist, researcher, or teacher), (e) fidelity assessment (yes/no), and (f) social validity assessment (yes/no). A one-sample Kolmogorov–Smirnov test indicated that Tau-BC values were not normally distributed (*p* < 0.001), supporting the use of non-parametric analyses. Differences in Tau-BC effect sizes across moderators were analyzed using the Mann–Whitney U test and the Kruskal–Wallis one-way analysis of variance (ANOVA). When the Kruskal–Wallis ANOVA test indicated significant differences, the Bonferroni post hoc test was conducted to control family-wise error. The significance level was set at α = 0.05.

## 3. Results

### 3.1. Characteristics of Studies

#### 3.1.1. Child Participants and Setting

As shown in Table 2, a total of 79 children participated in the 34 studies. Nine studies (26.5%) included children with comorbid disabilities, such as intellectual disabilities and spinal muscular atrophy (e.g., [19]; [20]). Various communication modes were targeted in implementing FCT: vocal (*n* = 11, 32.4%), picture (*n* = 4, 11.8%), gesture (*n* = 1, 2.9%), VOCA (*n* = 8, 23.5%), vocal and picture (*n* = 5, 14.7%), vocal and gesture (*n* = 2, 5.9%), or picture and VOCA (*n* = 3, 8.8%). The studies evaluated FCT in the home (*n* = 17, 50.0%), school (*n* = 12, 35.3%), living unit (*n* = 3, 8.8%), home and school (*n* = 1, 2.9%), or community (*n* = 1) settings.

#### 3.1.2. Target Challenging Behavior and Behavioral Functions

Target challenging behavior varied across studies. Eight studies (23.5%) focused on a single behavior, whereas the remaining twenty-six studies (76.5%) targeted multiple (comorbid) behaviors. Most studies included children engaging in aggression (*n* = 21, 61.8%), disruption (*n* = 14, 41.2%), SIB (*n* = 14, 41.3%), property destruction (*n* = 5, 14.7%), elopement (*n* = 4, 11.8%), tantrums (*n* = 5), or other behaviors (e.g., stereotypy, crying, screaming, and chin grinding). All studies involved children with challenging behavior maintained by a social reinforcement function, such as attention, escape, and tangible, with tangible being the most common function (*n* = 20, 58.8%). Children with multiple social reinforcement functions were included in 18 studies (52.9%). Only one study reported a non-social (automatic) reinforcement function for chin grinding behavior ([101]).

#### 3.1.3. Preference Assessment and FCT Components

Twelve studies (35.3%) conducted a preference assessment to identify potential reinforcers for conducting functional analysis or FCT design. Of these, four (17.6%) used a trial-based method (e.g., paired stimulus), three (8.8%) used free-operant, three (8.8%) used an indirect method, and one study (2.9%) used both indirect and trial-based approaches. Thirteen studies (38.2%) implemented FCT with additional intervention components, such as extinction (*n* = 6, 6.1%), delay-to-reinforcement (*n* = 3, 9.1%), tolerance training/reinforcement training *(n* = 1), extinction/delay-to-reinforcement (*n* = 1), lag reinforcement schedule (*n* = 1), minimizing antecedent stimuli (*n* = 1), and NCR (*n* = 1).

#### 3.1.4. Implementers and Dependent Variables

In most studies, the FCT intervention was implemented by researchers, therapists, or parents (or caregivers). Twelve studies (35.3%) involved parents or caregivers as implementers, either alone (*n* = 9, 26.5%), or in collaboration with a consultant, teacher, or researcher (*n* = 1 each). Primary dependent variables were challenging behavior and alternative FCR (see Table 2). In a few studies, additional dependent variables included appropriate behavior, such as task completion ([19]), compliance ([73]), and engagement ([38]).

### 3.2. Methodological Quality Assessment Results

#### 3.2.1. Study Design Quality

Results of the WWW Standards assessment revealed that only three studies (8.8%; [70]; [91]; [93]) met the WWC Standards without reservation. Seven studies (20.6%) met the Standards with reservations: six studies due to having fewer than five data points per condition while meeting all remaining criteria ([44]; [46]; [69]; [73]; [77]; [84]) and one due to residual treatment effects ([110]). Twenty-four studies (70.6%) were rated as not meeting the Standards.

#### 3.2.2. Assessment of Fidelity, Social Validity, Maintenance, and Generalization

Treatment fidelity was reported in fifteen studies (44.1%) and social validity was assessed in ten studies (29.4%). Only seven studies (20.6%) evaluated maintenance effects, and four studies (11.8%) reported generalization data.

#### 3.2.3. Information on Implementer Training and Implementation Support

Only seven studies (20.6%) provided information on the time required for initial training of interventionists; of these, three studies (8.8%) specified training durations of less than 1 h, and four (11.8%) reported training durations exceeding 1 hr. Two studies (5.9%) reported the provision of initial implementer training but did not specify the duration. Few studies provided information on whether any coaching or performance feedback was provided to parents, teachers, or therapists during FCT implementation. Among studies that involved parents, teachers, or therapists as FCT implementers, eight studies (23.5%) indicated that the frequency of coaching or performance feedback was daily (*n* = 5; 14.7%), weekly (*n* = 2; 5.5%), or across two sessions (*n* = 1 study; 3.0%). The limited reporting of training parameters did not permit analysis of their effects on intervention outcomes.

### 3.3. Effect Size and Moderators

Table 2 presents the effect sizes of FCT outcomes for each individual study, which ranged from questionable (≤0.65) to very effective ≥0.93). Table 3 presents aggregated Tau-BC by moderator category. The Tau-BC values ranged from very effective 0.93 (SE = 0.016) for FCT-only to effective 0.91 (SE = 0.032) for FCT combined with additional components. A Mann–Whitney test indicated no significant difference between FCT-only and FCT plus additional components. Similarly, Tau-BC values for outcome type ranged from effective 0.78 for replacement behavior to very effective 0.97 for challenging behavior, with no statistically significant difference across outcomes (*p* < 0.05).

Regarding setting, aggregated Tau-BC values were large for home (0.89), school (0.97), and community (0.94). A Kruskal–Wallis H test revealed a statistically significant difference in effect sizes across settings at *p* < 0.05 [H (2) = 6.239, *p* = 0.042]. Post hoc comparisons indicated a significant difference in mean rank scores between home and school, suggesting that FCT was more effective in school than home settings. The effect size for the community setting did not differ significantly from either home or school. For implementer type, Tau-BC values were large across all groups: therapists (0.99), teachers (0.95), researchers (0.92), and parents or caregivers (0.89), indicating moderately to highly effective. However, the Kruskal–Wallis H test revealed that the differences among the implementer groups were not statistically significant at *p* < 0.05 [H (3) = 2.875, *p* = 0.411]. For treatment fidelity, the Mann–Whitney test showed no statistically significant difference in effect size between studies with and without treatment fidelity assessment (U = 528.500, *p* = 0.263).

## 4. Discussion

This study synthesized 34 published SCD studies on FCT for young children with ASD, published prior to April 2022. Despite conducting our search without date restrictions, all included studies were published between 1996 and 2021, with none from before the late 1990s meeting our inclusion criteria. This temporal pattern likely reflects the evolution of FCT research since its introduction by [14] ([14]). The specific combination of our inclusion criteria (i.e., young children aged 1–8 years, implementation in natural settings, rigorous single-case designs, and graphical data suitable for effect size calculation) appears to characterize a later phase of FCT development. The marked increase in publications since 2011 may reflect continued growth in ASD prevalence estimates, with CDC surveillance data showing rates rising from 1 in 88 children in 2012 to 1 in 36 by 2020 ([15], [16]; [67]), along with corresponding federal initiatives emphasizing early intensive behavioral intervention and increased research funding priorities for evidence-based practices in natural settings ([113]; [97]). Although multiple systematic reviews and meta-analyses of FCT have examined the characteristics and methodological qualities of the FCT studies as well as effect sizes of the FCT intervention, this meta-analysis is the first to provide a summary of the FCT intervention outcomes for young children with ASD. The results of the analysis provide evidence of the benefits of FCT for this population, implications for practice, and suggestions that can inform future research. The main findings and discussions of the results are as follows.

### 4.1. Child and Setting Characteristics

The results revealed that most studies targeted preschool-aged children with ASD (3–5 years), consistent with [41] ([41]), who reviewed parent-implemented FCT for children with ASD, ages 2–14 years, and also found that the majority were between 3 and 5 years old. The predominance of preschool-aged participants (3–5 years) in our review has important implications for early intervention. This indicates that FCT has been successfully implemented during a critical developmental period when communication and behavioral patterns are most malleable. However, the limited inclusion of toddlers (<3 years) suggests potential missed opportunities for even earlier intervention, especially given that symptoms often emerge by the first birthday and diagnoses can be stable by 14 months ([81]; [83]). Although diagnostic timing varies based on symptom presentation and cognitive abilities ([2]; [56]), FCT adapted for pre-vocal communication modalities could be explored with younger toddlers showing early behavioral concerns. Early implementation during this critical window could diminish the necessity for more intensive support in the future by enabling individualized treatment strategies at the most opportune time.

While this review confirms the applicability of FCT during the preschool years, several findings highlight important implications for practice and research. The inclusion of children with comorbid conditions in some studies suggests that FCT can address the complex needs of young children with ASD who also present with intellectual disability, ADHD, physical impairments, or emotional difficulties. However, practitioners may need clearer guidance on adapting FCT procedures for such populations. Future research should examine how comorbidities influence FCT outcomes and develop implementation protocols that account for these complexities.

The range of behaviors targeted, including aggression, self-injury, property destruction, and elopement, which are commonly reported in the literature on challenging behavior in young children ([26]), demonstrates the broad applicability of FCT to the complex behavioral characteristics of young children with ASD, supporting its use as a first-line intervention when communication deficits contribute to challenging behavior, particularly for preschoolers ([28]). Furthermore, the various communication modalities used for the children (e.g., vocal, sign, picture, and VOCA) demonstrate the adaptability of FCT in accommodating different developmental levels and abilities. However, this variability may also indicate the need for guidelines to select the most appropriate communication modality for individual children. [103] ([103]) suggested that children with developmental disabilities are more likely to succeed using topography-based mands (e.g., sign) over selection-based mands (e.g., picture cards). FCT can also be more effective and efficient when communication modality aligns with a child’s motor, cognitive, and imitation abilities, as well as family preferences ([55]). Therefore, empirically based frameworks could assist practitioners in making such modality decisions rather than relying solely on clinical judgment.

We found that most studies targeting preschool-aged children were conducted in home or school settings, suggesting that the current literature provides limited information on outcomes of the FCT intervention for children with ASD served in community settings. This gap limits understanding of the effectiveness of FCT in more or less structured service environments. Future research should examine FCT implementation in community and inclusive settings to enhance ecological validity and support broader dissemination, with careful consideration of typical training requirements and resource constraints in these environments.

### 4.2. Preference Assessment

Although the researchers and practitioners have emphasized the benefits and importance of stimulus preference assessment for the assessment and intervention of challenging behavior in individuals with ASD or other developmental disabilities ([23]; [53]), the reporting rate for preference assessment was low (35.3%). Considering that identification of reinforcers is imperative when designing a function-based intervention, incorporating preference assessment into FCT should be standard practice when designing FCT interventions for young children with ASD, especially for behaviors maintained by social reinforcement functions, where identifying competing reinforcers is crucial for intervention success ([49]).

### 4.3. FCT Components, Intervention Duration, and Implementers

In examining the FCT components, we found that most studies (61.8%) evaluated the FCT alone without additional intervention components. When implementing FCT, it is critical to identify a replacement FCR that serves the same function as the challenging behavior and to teach the child to communicate for the attention, tangibles, or escape that they were previously requesting through challenging behavior. Although a continuous reinforcement schedule should be used during the initial training stage of the FCR to help the child learn the skills quickly, schedule thinning techniques, such as delay-to-reinforcement procedures, are often implemented to maintain appropriate rates of the communicative behavior and sustain the intervention implementation ([28]; [77]; [103]). However, in the current review, only five studies implemented FCT with delay-to-reinforcement or time delay to decrease the rate or density of reinforcement. [77] ([77]) suggested that schedule thinning procedures may be more effective for children with advanced communication skills. Similarly, FCT with extinction was implemented in only six studies, and these procedures were primarily delivered by therapists or researchers. This suggests that more research is needed on the use of FCT with schedule thinning procedures and extinction implemented by parents and teachers to examine whether these natural change agents can use FCT interventions effectively and to enhance and sustain the FCT implementation.

However, the type of FCR, whether a simple request or a more complex joint-attention behavior, may influence reinforcement effectiveness ([48]). While delayed reinforcement can facilitate schedule thinning, extended delays may risk extinction of the FCR or the resurgence of maladaptive behaviors ([50]; [92]; [37]). Recent research has advanced schedule thinning methodologies, including enhanced multiple schedule procedures ([12]; [47]), integration of competing stimuli ([40]), and alternative thinning approaches ([60]). Although multiple schedules can establish stimulus control over FCR, their implementation may require contrived discriminative stimuli that are not naturally present in typical environments ([9]; [88]). In addition, many FCT protocols involve criterion shaping of FCRs ([42]); however, concurrent procedures are often underreported ([88]). The systematic reporting of these procedures and careful consideration of the communication target and reinforcement schedule would enhance comparability across studies and inform more effective FCT implementation, particularly by natural change agents.

### 4.4. Methodological Quality

Our results indicated that only 10 studies (29.4%) on FCT for young children with ASD met the WWC Standards for design quality without reservation (8.8%) or with reservations (20.6%). In contrast, [17] ([17]) found that more than half of the studies on FCT they reviewed (*n* = 44) met the WWC Standards, whereas [79] ([79]) found that 77% of the included studies (*N* = 37) met WWC Standards with or without reservation. [66] ([66]) systematically reviewed SCD studies on FCT in early care and education settings and reported that none of the studies included in the analysis (*N* = 20) met the WWC Standards without reservation, and 40% met the standards with reservation.

The primary reason for not satisfying the WWC Standards was insufficient data points per phase. In synthesizing 20 studies on FCT for young children in early care and education settings, [66] ([66]) reported that 0% of studies met the WWC Standards without reservation and 40% met the standard with reservation. These results contrast with [111] ([111]), who reported that over 70% of 17 school-based FCT studies met WWC Standards (30% without and 42% with reservation). Differences in participant characteristics, intervention settings, and reporting rigor likely account for the discrepancies. Nonetheless, the relatively small proportion of studies meeting WWC Standards in the present review stress the need for greater methodological rigor in FCT research with young children with ASD. However, it should be noted that WWC Standards have important limitations, including the exclusion of visual analysis for evaluating internal validity and potential oversight of methodological issues that visual analysis would detect ([61]). Despite the limitations inherent in the WWC Standards, the consistent finding across reviews that many FCT studies include an insufficient number of data points per phase highlights key areas for methodological improvement, particularly when data patterns fail to show clear differentiation between baseline and treatment phases. In addition to ensuring adequate data points, future researchers should integrate rigorous design standards with comprehensive visual analysis to strengthen internal validity and enhance the interpretability of study findings.

We also examined the reporting rates for implementer training procedures, treatment integrity, and social validity, as well as the rates for intervention maintenance and generalization evaluations. We found that only seven studies (20.6%) provided information on the time required for initial implementer training, and only seven out of the eight studies provided information on the frequency of coaching or feedback. These limited training data are concerning, given the critical importance of thorough training provision ([72]), the increased risk of challenging behaviors when interventions are implemented by inadequately trained individuals ([72]), and the association between classroom staff fidelity improvement with coaching support and the marked reductions in challenging behavior ([45]). Furthermore, only 44.1% of the studies reported on treatment fidelity, which corroborates findings of previous review studies on FCT for children with disabilities including ASD ([1]; [41]; [66]). Future research should carefully plan initial training and coaching procedures for natural change agents based on baseline observations, as inadequate initial training may contribute to ineffective implementation and maintenance issues ([3]; [24]). Investigating telehealth and distance-based training approaches could help address constraints in time, resources, and accessibility while supporting skill generalization and maintenance across home, school, and community settings ([21]). Such approaches should be evaluated for parents, therapists, and other natural change agents to optimize intervention fidelity and long-term outcomes ([106]).

We also found that despite the continuing emphasis on social validity in behavioral interventions for young children ([63]; [82]), the rate for reporting social validity assessment in the current FCT literature remained low (29.4%). This reporting rate was smaller than that reported by [41] ([41]), who found that about half of studies they reviewed included social validity assessments. We also found that among studies that reported social validity, the assessments were typically conducted as one-time, post-treatment questionnaires. As noted in the literature, such measures can sometimes overestimate satisfaction and may function more as an indirect measure of perceived effectiveness rather than a comprehensive evaluation of whether the magnitude of behavior change was sufficient to produce a meaningful outcome in the child’s or family’s quality of life ([51]). Given that implementing socially valid interventions is essential for improving outcomes related to challenging behavior in young children ([36]; [72]; [82]), the results suggest that future studies on FCT for young children with ASD should not only consistently assess social validity, but also employ more robust, longitudinal methods that assess the pragmatic significance of outcomes and their sustained impact. Furthermore, it would be valuable for future research to examine the role of technology in FCT delivery, including the assessment of social validity in implementer training and coaching via telehealth, given its increasing relevance in the implementation of behavioral interventions ([18]; [104]).

The results also indicate that the current literature on FCT for young children with ASD provides limited information on the maintenance (20.6% of studies) and generalization (11.8% of studies) of the intervention, a finding consistent with previous reviews ([28]; [79]). This limitation may reflect not only a limited number of studies but also a lack of depth. The literature indicates that there is minimal research comparing reinforcement-thinning methods, such as multiple schedules, contingency-based delays, and demand fading, to determine which approaches best support generalized outcomes ([5]; [43]). Considering that generalization assessments are often methodologically limited, future research should not only evaluate generalization more frequently but do so with greater rigor, such as by (a) using experimental designs that compare behavior across contexts before and after treatment to address the potential absence of an establishing operation ([5]) and (b) examining the generalization of both direct effects (reductions in challenging behavior) and indirect effects (increases in alternative skills). These gaps have direct practical implications. They highlight the importance of incorporating maintenance and generalization supports into FCT intervention implementation for young children with ASD. Although reinforcing every FCR may be necessary early in treatment to establish the communicative response and reduce challenging behavior, sustaining behavior change may require planned maintenance and generalization promotion strategies. Researchers should assist parents, teachers, and therapists in using maintenance and generalization strategies such as schedule thinning ([47]; [77]) and incorporating common stimuli across settings ([33]; [79]). Furthermore, as discussed above, providing systematic training and ongoing coaching for these natural change agents are essential to ensure consistent and effective implementation across settings.

### 4.5. Magnitude of FCT Intervention and Moderating Variables

The effect size analysis results indicate that FCT is very effective for reducing challenging behavior (Tau-BC = 0.97) and effective for replacement behavior (Tau-BC = 0.78). These effect sizes were larger than those reported in previous meta-analyses of FCT studies. [17] ([17]) reported an overall Tau-U effect size of 0.68 for challenging behavior and 0.65 for FCR. The larger effect sizes in our current review may be partly attributed to the participants’ age range and types of disability, as the present analysis focused exclusively on young children aged 2 to 8 years with ASD. In Chezan et al., although the overall effect size of FCT was lower, the effect size for preschool-aged children was greater than those for school-aged children and adults. [52] ([52]) also found that the effect sizes of FCT were larger for young children, particularly those with ASD, compared to older children and adults. It is also important to consider that the presence of comorbid intellectual disabilities or variations in autism severity (i.e., Level 1 to Level 3) may moderate treatment outcomes. Children with greater cognitive or adaptive challenges may require more intensive or prolonged FCT to achieve comparable gains, whereas those with higher functioning levels may acquire communicative alternatives more rapidly. However, most of the reviewed studies did not clearly report participants’ cognitive functioning or autism severity, which limited our ability to examine their potential moderating effects. Future research should report this information to better understand how these characteristics influence FCT effectiveness and to inform individualized intervention planning.

Although studies that evaluated FCT alone demonstrated a slightly larger effect size than studies with FCT plus additional components, the difference was not statistically significant. The results also indicate that the intervention setting has moderating effects. The overall effect size for school settings was significantly larger than that of home settings. The therapists and researchers were the most common implementers of FCT; however, several studies trained parents or caregivers to implement the intervention. The proportion of parent or caregiver implementers was higher than reported in a previous review of studies on FCT ([68]) and similar to findings for FCT studies with young children ([28]). Although statistically significant differences were found among implementers, the smallest effect size was found when parents or caregivers served as implementers. These smaller effect sizes for home settings and parents or caregiver implementers may suggest a need for adaptations of FCT intervention to better fit family needs and routines in home environments, where competing demands and limited resources may affect implementation fidelity ([28]).

The limited reporting of implementer training details represents a critical gap in the literature. Although examining training effects was not the primary focus of this review, this lack of specificity makes it difficult to determine the appropriate training procedures and dosage needed to achieve high implementation fidelity. Access to empirically validated training, such as behavioral skills training ([114]), is particularly important for supporting natural change agents in implementing FCT. Contextual factors, such as distractions or unexpected changes in client behavior, may still disrupt performance ([4]; [76]). To address these gaps, we recommend that future research clearly report (a) the duration and format of initial training, (b) competency criteria required before independent implementation, (c) the frequency and type of ongoing support, and (d) the relationship between training dosage and treatment fidelity.

### 4.6. Limitations and Conclusion

Limitations of the present meta-analysis should be considered when interpreting the results. Despite various efforts to identify articles to include in the analysis, such as searching multiple databases and examining references of existing review papers, there may be additional relevant articles that were not found. A second limitation is that although it is important to examine effect sizes and moderator effects for maintenance and generalization outcomes of FCT interventions, the small number of studies reporting these prevented the calculation of effect size estimates. Results of the current meta-analysis clearly indicate that relatively little is known about the extent of the maintenance and generalization effects of FCT intervention on behaviors of young children with ASD. Therefore, we suggest that future researchers investigating FCT for children with ASD actively evaluate maintenance and generalization effects. An accumulation of these studies would allow future researchers to examine potential moderators of maintenance and generalization effects.

This study further supports previous findings that FCT is an evidence-based practice for young children with ASD. Most of the reviewed studies showed strong effects on decreasing challenging behavior and increasing replacement behavior. However, methodological quality should be improved to strengthen this evidence base. Given the significant moderating effect of setting on child behavior, indicating better outcomes when implemented at school than at home, future researchers should consider implementing and examining effective parent training and implementation support strategies (e.g., coaching and feedback) and assessing the competency of implementers in delivering FCT with fidelity.

While information on intervention duration, intensity, and comprehensive training procedures would provide valuable insights for clinical implementation, we did not examine these variables as they fell outside our primary research objectives. Furthermore, insufficient reporting across studies prevented meaningful analyses of these variables. Future meta-analyses specifically designed to examine dosage–response relationships in FCT would be valuable for guiding evidence-based implementation.

## Figures and Tables

**Figure 1 behavsci-15-01688-f001:**
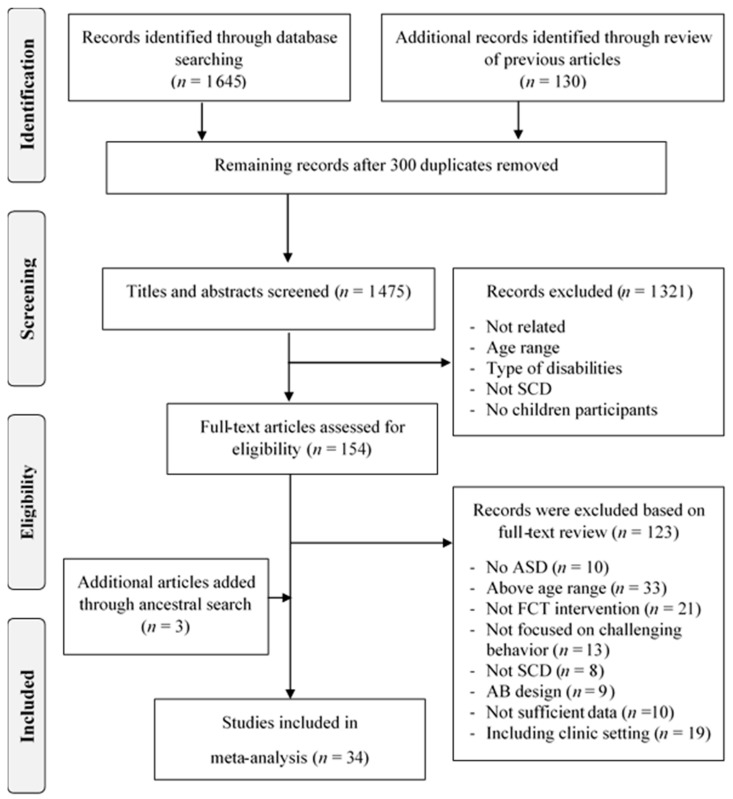
Flow chart of the study selection process.

**Table 1 behavsci-15-01688-t001:** Coding variables.

Category	Coding Variables
Child characteristics	(a) the number of participants (b) age (in months) (c) diagnosis (d) communication mode
Communication mode	(a) vocal (b) picture (c) sign (d) voice output communication aid (VOCA)
FCT features	(a) target challenging behavior (b) function of challenging behavior (c) setting (d) preference assessment (e) FCT components (f) implementer (g) implementer training (length of initial training and frequency of coaching or feedback)
Target challenging behavior	(a) aggression (b) disruption (c) property destruction (d) tantrum (e) self-injury (f) stereotypy (g) elopement (h) crying, screaming (i) chin grinding
Function of challenging behavior	(a) social (attention, tangible, escape) (b) automatic (non-social)

**Table 2 behavsci-15-01688-t002:** Characteristics of individual studies.

Author (Year)	*n*	Age	Diag.	Comm. Mode	Target CB	Function	Pref. Ass.	FCT Compo.	Setting	Implementer	Initial Training	Coach./Feed.	FD	SV	M	G	DV	Tau-BC	WWC Stand.
[8] ([8])	1	5	ASD	Picture	SIB	Social (A)	NR	FCT + TT/ST	Home	Mother	NR	NR	Y	N	N	N	CB RB	1.0 0.89	0
[11] ([11])	1	7	ASD/ID	Vocal	SIB, Aggression	Social (T, E)	NR	FCT + DR	School	Therapist	NR	NR	N	N	N	N	CB RB	1.0 1.0	0
[13] ([13])	1	7	ASD	Picture	Aggression	Social (T)	NR	FCT + Ext	School	Therapist	NR	NR	N	N	N	N	CB RB	0.92 NA	0
[19] ([19])	2	5.2, 6.2	SMA/PDD, ASD/ID	Picture, VOCA	SIB, Aggression, PD	Social (E/T, A/E)	T	FCT	Home	Mother	NR	NR	N	N	N	N	CB RB	0.67 0.83	0
[20] ([20])	1	6	ASD/ID	Vocal	Aggression, Stereotypy	Social (T)	T	FCT	School	Researcher	NR	NR	N	N	N	N	CB RB	1.0 0.75	0
[22] ([22])	1	4	PDD-NOS	Vocal	Disruption	Social (T)	NR	FCT	School	Researcher	NR	NR	N	N	N	N	CB RB	1.0 1.0	0
[30] ([30])	1	5	ASD	Gesture	Elopement. Disruption	Social (T)	NR	FCT + DR	Com.	Therapist	NR	NR	N	N	N	N	CB RB	0.59 NA	0
[31] ([31])	2	8	ASD (Asperger)	Vocal	Aggression, PD, SIB, Disruption	Social (T)	NR	FCT	School	Researcher	NR	NR	Y	N	N	N	CB RB	0.97 0.85	0
[32] ([32])	2	2.5, 4.5	ASD	VOCA	SIB, Disruption, Aggression	Social (A/T, E)	F	FCT	Home	Mother/ Researcher	NR	NR	N	N	N	N	CB RB	−0.89 NA	0
[34] ([34])	1	8	ASD	Vocal	Disruption	Social (A/T, E)	NR	FCT	Home	Therapist	NR	NR	N	N	N	N	CB RB	0.87 NA	0
[38] ([38])	1	7.5	ASD	VOCA	Disruption	Social (A/T; E)	NR	FCT	School	Therapist	NR	NR	Y	N	Y	N	CB RB	1.0 1.0	0
[39] ([39])	1	4	ASD	VOCA	Aggression	Social (T)	NR	FCT	School	Teacher	NR	NR	N	N	N	N	CB RB	0.85 1.0	0
[44] ([44])	1	4	ASD	Vocal	Elopement	Social (T)	F	FCT	School	Teacher	<1 h	Daily	Y	Y	N	N	CB RB	1.0 NA	1
[46] ([46])	1	8	ASD	Vocal	Aggression, Disruption	Social (A, E)	NR	FCT + Ext	Living Unit	Therapist	NR	NR	N	N	N	N	CB RB	0.49 0.79	1
[57] ([57])	1	7	ASD/ ID	Picture/ VOCA	Aggression, SIB, PD	Social (T)	T	FCT	Living unit	Therapist	NR	NR	N	N	N	N	CB RB	0.98 NA	0
[59] ([59])	3	4, 4, 5	ASD	Vocal	Aggression	Social (E)	NR	FCT + AI	Home	Parents	NR	Daily	N	Y	Y	N	CB	0.98 N/A	0
[69] ([69])	1	4	PDD	Picture/ Vocal	Tantrum	Social (T/A)	T	FCT	Home	Researcher	NR	NR	N	N	N	N	CB RB	1.0 1.0	2
[70] ([70])	3	7.9, 4.1, 4.8	ASD	Picture, Vocal	Aggression, SIB, Tantrum	Social (T)	T	FCT	Home	Parents	Y	NR	Y	Y	Y	Y	CB RB	0.1.0 0.49	1
[73] ([73])	1	4	ASD	Vocal	Aggression, Disruption	Social (E)	NR	FCT + NCR	Home	Researcher	NR	NR	Y	N	N	N	CB RB	1.0 0.53	1
[77] ([77])	2	5, 6	ASD	VOCA	Disruption, SIB	Social (A)	I	FCT + Ext	School	Researcher	NR	NR	Y	N	N	N	CB	1.0 NA	1
[78] ([78])	2	5, 8	ASD, ASD+ID	Picture, Vocal	Disruption, SIB	Social (A)	I, T	FCT + Ext	School	Researcher	NR	NR	Y	Y	Y	Y	CB RB	0.82 0.61	0
[80] ([80])	1	4	ASD	VOCA	Aggression Elopement	Social (A)	NR	FCT + Ext	Home	Mother	Y	Daily	Y	Y	N	N	CB RB	1.0 0.78	0
[84] ([84])	2	3, 5	ASD	Picture, Vocal	Aggression, Disruption	Social (A, E)	I	FCT + Ext/DR	Home	Researcher	NR	NR	Y	Y	N	N	CB RB	1.0 1.0	1
[90] ([90])	2	2.8, 6.9	ASD	VOCA	SIB, Aggression, PD	Social (T; T/E)	NR	FCT	Home	Mother	<1 h	Weekly	Y	N	N	N	CB RB	0.00 NA	0
[91] ([91])	3	4, 4, 5	ASD	Gesture, Vocal	Aggression, Tantrums	Social (T, E)	I	FCT	School/ Home	Parents/ Teacher	>1 h	2 SS	Y	N	N	N	CB RB	0.86 0.90	2
[93] ([93])	2	8, 8	ASD	Gesture, Vocal	Aggression, SIB, Disruption	Social (A/T)	NR	FCT	School	Teacher	NR	NR	N	N	Y	N	CB RB	1.0 0.65	2
[99] ([99])	1	4	ASD	Vocal	Tantrums	Social (T)	NR	FCT	Home	Teacher	NR	NR	N	Y	N	N	CB RB	1.0 −0.04	0
[94] ([94])	2	4, 5	ASD, ASD+ ADHD	VOCA	Crying, Elopement, Aggression	Social (A, E)	F	FCT + LS	Home	Researcher	NR	NR	N	N	N	N	CB RB	0.98 −0.09	0
[95] ([95])	2	3, 4	ASD	Picture	Tantrums	Social (A, E)	NR	FCT	Home	Parents	>1 h	NR	Y	N	N	N	CB RB	0.88 0.90	0
[100] ([100])	4	6, 3, 6, 4	ASD	VOCA	Aggression, PD, SIB, Screaming	Social (A, E)	NR	FCT	Home	Consultant + Caregiver	NR	NR	N	N	Y	Y	CB RB	0.98 N/A	0
[105] ([105])	10	3–8	ASD	Picture, Vocal	Multiple Individually Defined	Social (A, E)	NR	FCT	Home	Mother	<1 h	Daily	Y	Y	N	Y	CB RB	0.97 N/A	0
[101] ([101])	1	7	PDD-NOS, ID	Picture	Aggression, Chin Grinding	Automatic Social (A)	NR	FCT+ Ext	Living Unit	Therapist	NR	NR	N	N	N	N	CB RB	0.88 NA	0
[109] ([109])	17	2.4–6.7	PDD-NOS (10), ASD (7)	Vocal	Aggression, PD, SIB, Disruption	Social (A, E, T, E/T)	NR	FCT	Home	Parents	>1 h	Weekly	N	Y	N	N	CB RB	0.61 NA	0
[110] ([110])	2	7, 7	ASD+ID, ASD+DD	VOCA, Picture	SIB, Disruption	Social (E)	NR	FCT	School	P-educator	<1 h	Daily	Y	Y	Y	N	CB RB	0.70 NA	1

Note. A = attention; AI = antecedent intervention; I = indirect; CB = challenging behavior; Comm. = communication; Coach. = coaching; Com = community; Compo. = components; DBD = disruptive behavior disorder; D = direct; Diag. = diagnosis; DR = delay of reinforcement; E = escape; Ext = extinction; F = free-operant; FCR = functional communication response; FCT = functional communication training; FD = fidelity; Feed. = feedback; G = generalization; I = indirect; ID = intellectual disability; LS = lag schedule; M = maintenance; Mod. = moderate; *n* = number of child participants; N = no; Noncom. = noncompliance; NCR = noncontingent reinforcement; NR = not reported; NS = not specified; P = para; Pref. = preference; PD = property destruction; PDD-NOS = Pervasive Developmental Disorder-Not Otherwise Specified; RB: replacement behavior; S. = severe; SMA = spinal muscular atrophy; SS = sessions; ST = schedule thinning; Stand. = standards; SV = social validity; T = trial-based; T = tangible, TT = Tolerance Training; VOCA = voice output communication aid; WWC Standards (2 = meet WWC SCD standards without reservations; 1 = meet WWC SCD standards with reservations; 0 = does not meet WWC SCD standards); Y = yes.

**Table 3 behavsci-15-01688-t003:** Moderator effects and differences among moderators.

Moderator	*k*	Tau-BC (SE)	H/U	*p*
FCT component				
FCT-only	50	0.93 (0.016)	479.500	0.561
FCT plus additional component	21	0.91 (0.032)		
Outcome				
Challenging behavior	44	0.97 (0.007)	437.000	0.356
Replacement behavior	23	0.78 (0.065)		
Setting				
Home ^a^	36	0.89 (0.029)	6.329	0.042
School ^b^	23	0.97 (0.015)		
Community ^ab^	4	0.94 (0.065)		
Implementer				
Parent or caregiver	20	0.89 (0.039)	2.875	0.411
Therapist	12	0.99 (0.110)		
Researcher	26	0.92 (0.029)		
Teacher	6	0.95 (0.044)		
Fidelity assessment				
Yes	39	0.95 (0.009)	528.500	0.263
No	32	0.98 (0.009)		
Social validity assessment				
Yes	55	0.89 (0.046)	378.000	0.368
No	16	0.93 (0.016)		

Note. FCT = functional communication training; k = number of contrasts; SE = standard error. Effect sizes with different superscript letters within a moderator are significantly different at the *p* < 0.05; home (a) and school (b) have different superscript letters, whereas both home (a) and school (b) share superscript letters with community (ab), indicating that the effect size for community is not significantly different from either home or school.

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
