# Peer review of "A Meta-Analysis of Functional Communication Training for Young Children with ASD and Challenging Behavior in Natural Settings"

_behavsci, 2025, doi:10.3390/bs15121688_

Round 1

Reviewer 1 Report

Comments and Suggestions for Authors

Overall I really enjoyed the paper and especially thought the results were well done.  There is emphasis on where the data is lacking and why that is relevant for the implementation as well as how that is relevant to clinical practice. 

There are some sections where the grammar and sentence structure gave me pause and I had to read two or three times to understand. Please look at some of these to make clearer:

lines 31-35, 64-65, 76-79, 355-359

Take a look at 41-49.  There appear to be too many sentences that say the same thing.  The first introduction paragraph may benefit from some work to make more concise and to the point. 

Table 1 is very difficult to look at in its current form.  The columns and rows run into each other, and it is difficult to read.  I would consider seeing if there is a way, within the journal's parameters, that you could include the needed data but make this easier to follow.  

The discussion is the strength of the paper laying out both the clinical use of these studies and the need for the exact kind of studies that need to come in the future. 

Author Response

We would like to thank the reviewers for their thoughtful comments and helpful suggestions. We have carefully revised the manuscript accordingly and believe the changes have improved clarity and rigor. In the revised manuscript, major revisions are highlighted in blue. Below we provide a point-by-point response to each comment.

Reviewer 2 Report

Comments and Suggestions for Authors

Review of the Meta-Analysis on Functional Communication Training (FCT) for Children with Autism

Many thanks to the editor for inviting me to review this meta-analysis on FCT for children with autism. I was very impressed by the clarity and simplicity of the work, and I will try to suggest improvements to make it more technically robust and comprehensive.

Abstract
The abstract is very clear and accessible; however, it may be a little too concise. Adding a few more details on the methodology or the main results could enhance its scientific value without compromising readability.

Introduction
This section is also clearly written, but I suggest some important improvements.
- When discussing the prevalence of the problem, it is necessary to specify the reference country each time, since prevalence rates can vary depending on culture and sampling methods.
- The authors do not sufficiently address the literature on functional analysis, taking the functions of problem behaviors for granted. This section should provide a clearer and more complete overview of how antecedents of behaviors are identified and how this process relates to FCT-based interventions.
- In addition to discussing studies on the functional analysis and effectiveness of FCT, the introduction should also explore precursors and moderators that may influence behavioral intervention outcomes.

Method
- Please clarify whether the keyword searches were conducted in the title, abstract, or full text. This information is crucial to make the search strategy replicable.
- I strongly recommend including, in an appendix, a table listing all 154 full-text articles that were screened, with the specific reasons for exclusion. This will enhance the transparency and replicability of the review process.
- The description of stimulus preference assessments is too limited. The authors treat this variable as dichotomous, whereas in practice, preference assessments can be indirect, trial-based, or free-operant.
  Moreover, trial-based preference assessments could act as important precursors influencing intervention outcomes.
- Around line 179, where the term “reinforcement thinning” is mentioned, it would be helpful to clarify that this refers to schedule thinning procedures.
- From line 180 onwards, when referring to “averages,” it is unclear how these variables were calculated, particularly if they are dichotomous (e.g., “mastered/not mastered”). Please specify the computational methods used.
- Similarly, when discussing methodological quality (line 194), please provide examples of the instruments or checklists employed, along with sample items or response formats.
- In the section on moderator analyses, it is unclear whether the assumptions for non-parametric analyses were checked. Please address this.
- The variable “communication modality” appears in the literature but seems to be missing from the moderator analyses; consider including or explaining its exclusion.

Results
- There appears to be a repetition around line 251, where a variable already described earlier is restated descriptively.
- The authors treat target problem behaviors as isolated, whereas Table 1 suggests that behaviors often co-occur. Please report the percentages of comorbid behaviors and their respective antecedents (e.g., attention, escape).
- Coding for stimulus preference assessments appears to be dichotomous again; this should be clarified.
- In the methodology and study quality section (lines 280–288), I strongly recommend including a table listing the quality assessment items and the corresponding responses for each study, along with the summary scores.
- A broader issue across the manuscript is the lack of information on duration, dosage, and intensity of the interventions, as well as the training procedures for implementers. Note that the implementers are often RBTs, which may influence both training and intervention outcomes.
- Extend the notes under the effect size table (lines 342–343) to improve clarity.

Discussion
One of the main issues is that the discussion section focuses too much on restating results, rather than interpreting their clinical or practical implications.
- When intellectual disability is present as a moderator or comorbidity, discuss how it may have influenced treatment outcomes. Similarly, consider how varying levels of autism severity (from Level 1 to Level 3) might act as moderators. These could also be discussed under the limitations.
- When referring to reinforcement schedules and other associated procedures, it would be useful to consider whether the communicative target was a simple request or a more complex joint-attention response.
- Lines 397–399: Please address that many FCT protocols involve criterion shaping, and that concurrent behavioral procedures (e.g., extinction or non-contingent reinforcement) are often underreported. Clarifying this could guide future studies and promote more standardized reporting.
- In the methodological quality discussion, I encourage the authors to offer specific recommendations for improving study design in future research (e.g., at line 411).
- Around line 430, it would be interesting to analyze training procedures for families or therapists before intervention implementation, not only post-intervention, as this may relate to maintenance issues.
- The discussion could also address how telehealth and distance training may improve generalization and maintenance of skills across settings, including community environments.
- Finally, future studies should include measures of social validity and assess the role of technology more explicitly, given its increasing relevance in current FCT research.

Conclusion
I sincerely thank the editor and the authors for considering these suggestions to strengthen the manuscript. The study is a valuable contribution to the literature and should be considered for publication after the recommended revisions are made.

Good luck with the revision process.

An Italian similar SCD

Caracciolo, S., Esposito, M., Dipierro, M. T., Smith, D., & Martella, N. (2020). Functional Communication Training for Autism: An Italian Case Study. Int J Autism & Relat Disabil: IJARD-133. DOI, 10, 2642-3227.

Author Response

(The authors gave the same response as above.)

Reviewer 3 Report

Comments and Suggestions for Authors

Dear Authors, I congratulate for the work undertaken conceiving the paper "A Meta-Analysis of Functional Communication Training for Young Children with ASD and Challenging Behavior in Natural Settings", which addresses the effectiveness of single-case design studies involving functional communication training (FCT) for children aged 2 to 8 with autism spectrum disorder (ASD), upon which inadequacies in reporting key factors like preference assessment and treatment fidelity were noted, although notably large effects have been observed.

Besides the theoretical framework that could be delved, there are some other issues regarding to methodology and results reporting that I would like call your attention for, which are detailed in report attached. I also would like to stress my comments are made in a constructively perspective, aiming the improvement of your work.

Best regards,

Author Response

(The authors gave the same response as above.)

Round 2

Reviewer 3 Report

Comments and Suggestions for Authors

Dear Authors, let me congratulate for resilient work undertaken in revising the manuscript, which went through some improvements from the first version. There are though some other issues that I would like to call your attention for, which are detailed in my report attached. Some regard methodological issues, whereas others concern the way some topics and their roots/implications were discussed and could be broadly delved, even when literature somehow limits a concret analysis on them.

For instance, the scarcity of studies reporting FCT implementer training procedures should not have to prevent the discussion on the contexts by which trainees must go through to develop skills that become them able to better serve their patients.

Best regards, 

Author Response

Dear Reviewer 

We would like to thank the reviewers for their thoughtful comments and helpful suggestions. We have carefully revised the manuscript accordingly and believe the changes have improved clarity and rigor. In the revised manuscript, major revisions are highlighted in blue. Below we provide a point-by-point response to each comment.

Sincerely yours, 

Authors
